# Increased PIEZO1 Expression Is Associated with Worse Clinical Outcomes in Hormone-Receptor-Negative Breast Cancer Patients

**DOI:** 10.3390/cancers16040683

**Published:** 2024-02-06

**Authors:** Rylee Ann Poole, Qingfei Wang, Alo Ray, Kazuaki Takabe, Mateusz Opyrchal, Eriko Katsuta

**Affiliations:** 1Division of Hematology/Oncology, Indiana University, Indianapolis, IN 46202, USA; rypoole@iu.edu (R.A.P.);; 2Indiana University Melvin and Bren Simon Comprehensive Cancer Center, Indianapolis, IN 46202, USA; 3Department of Surgical Oncology, Roswell Park Comprehensive Cancer Center, Buffalo, NY 14203, USA; 4Department of Oncology, Graduate School of Medicine, Yokohama City University, Yokohama 236-0004, Japan

**Keywords:** ion channels, hormone-receptor-negative breast cancer, PIEZO, mechano-signaling

## Abstract

**Simple Summary:**

PIEZO1 is a mechanically activated ion channel connected to many important cellular functions. While implicated to various degrees in different types of cancers, the clinical significance of PIEZO1 has not been explored in breast cancer. We conducted various bioinformatic analyses on PIEZO1 in breast cancer, using publicly available online datasets from The Cancer Genome Atlas and GSE3494. Our results show that PIEZO1 expression is higher in hormone-receptor (HR)-negative cohorts than HR-positive cohorts. We also found that high PIEZO1 expression is correlated with worse survival outcomes in HR-negative patients, suggesting that PIEZO1 could be utilized as a prognostic biomarker in HR-negative breast cancer. Further analysis suggests that these worse survival outcomes may be due to increased aggressive cancer pathways, including epithelial–mesenchymal transition and hypoxia, along with decreased CD8+ and CD4+ T cell infiltration in high-PIEZO1 HR-negative tumors.

**Abstract:**

PIEZO1 plays a crucial role in the human body as a mechanosensory ion channel. It has been demonstrated that PIEZO1 is important in tissue development and regulating many essential physiological processes. Studies have suggested that the PIEZO1 ion channel plays a role in invasion and progression in cancer; elevated levels of PIEZO1 have been correlated with increased migration in breast cancer cells, chemo-resistance and invasion in gastric cancer cells, and increased invasion of osteosarcoma cells. In addition, high PIEZO1 expression levels were correlated with a worse prognosis in glioma patients. On the other hand, studies in lung cancer have attributed high PIEZO1 levels to better patient outcomes. However, the clinical impact of PIEZO1 in breast cancer is not well characterized. Therefore, our goal was to determine the clinical relevance of PIEZO1 in breast cancer. An analysis of breast cancer data from The Cancer Genome Atlas (TCGA) was conducted to investigate PIEZO1 expression levels and correlation to survival, followed by validation in an independent dataset, GSE3494. We also performed gene set enrichment analysis (GSEA) and pathway enrichment analysis. We also analyzed the immune cell composition in breast tumors from TCGA through a CIBERSORT algorithm. Our results demonstrated that the PIEZO1 expression levels are higher in hormone-receptor (HR)-negative than in HR-positive cohorts. High PIEZO1 expression is correlated with a significant decrease in survival in HR-negative cohorts, especially in triple-negative breast cancer (TNBC), suggesting that PIEZO1 could be utilized as a prognostic biomarker in HR-negative breast cancer. GSEA showed that various signaling pathways associated with more invasive phenotypes and resistance to treatments, including epithelial–mesenchymal transition (EMT), hypoxia, and multiple signaling pathways, are enriched in high-PIEZO1 HR-negative tumors. Our results also demonstrated a decrease in CD8+ and CD4+ T cell infiltration in high-PIEZO1 HR-negative tumors. Further investigations are necessary to elucidate the mechanistic roles of PIEZO1 in HR-negative breast cancer.

## 1. Introduction

Breast cancer is the most abundant type of cancer globally, accounting for 12% of all cancer diagnoses in 2021 [1]. As breast cancer continues to impact millions of people worldwide, it is our goal to identify novel biomarkers and therapeutic targets to reduce breast cancer morbidity and mortality. In recent years, researchers have become increasingly interested in how mechanical forces within the tumor microenvironment (TME), including tension, shear stress, and other forces, impact the survival and propagation of cancer cells [2]. Mechanotransduction, or the sensation of mechanical stimuli and subsequent conversion to biochemical signals, occurs through various adhesion molecules, ion channels, and cytoskeletal components. Mechanosensation is essential for the regulation of many downstream signaling pathways, many of which are implicated in cancer [3]. A key set of proteins involved in mechanosensation is the PIEZO proteins [4].

PIEZO proteins are evolutionarily conserved, mechanically activated ion channels [4,5,6,7]. The PIEZO family is composed of two members—PIEZO1 and PIEZO2 [6]. PIEZO1 is primarily expressed in non-excitable cell types, while PIEZO2 confers mechanosensory abilities to excitable cell types such as sensory neurons and Merkel cells [8]. PIEZO1 plays major roles in sensing various forces, including shear force, tension, and stretching, and then transmitting these signals to stimulate downstream pathways important for development and homeostasis in both normal physiology and disease. PIEZO1 has been demonstrated to play critical roles in endothelial cell organization [9,10], vascular patterning and development [9,10], neural cell differentiation [9], bone formation and homeostasis [11,12], and the differentiation of various cell types [8].

There has been a growing interest in exploring PIEZO proteins and their mechanosensory abilities in the context of malignancies and the TME. Recently, we discovered that increased expression levels of PIEZO2 are correlated with worse survival outcomes in triple-negative breast cancer (TNBC) through increased AKT activation, stabilized SNAIL, and the repression of E-cadherin transcription [13]. PIEZO1 expression has also been demonstrated to contribute to various physiological processes within the hallmarks of cancer, including epithelial–mesenchymal transition (EMT) [14,15], angiogenesis [15,16,17], hypoxia [15,16,18], metabolic alteration [19], inflammation [18], and the functional gain or loss of various signaling pathways [14,16,20,21,22]. The involvement of PIEZO1 within these processes and conditions results in discernable effects on cancer progression, as high PIEZO1 expression has been shown to correlate with increased migration and chemo-resistance in gastric cancer cells [23,24] and the increased invasion of osteosarcoma cells [25]. In breast cancer, Li et al. showed that the pharmacological inhibition of PIEZO1 led to the decreased motility of MCF-7 breast cancer cells [26], while Yu et al. reported that PIEZO1 knockdown in MDA-MB-231 breast cancer cells led to increased unconfined cell migration [27]. In addition, it is well known that these aforementioned pro-cancer processes impact the immune population of the TME, either by immune suppression or immune exclusion. For example, EMT, hypoxia, and TGF-β signaling have all been shown to be inversely correlated with CD8+ and/or CD4+ T cell infiltration in various cancer types [28,29,30]. CD8+ T cells are major drivers of the anti-cancer immune response, with decreased levels of CD8+ infiltration correlating with worse survival outcomes in breast cancer patients [31,32]. Therefore, PIEZO1 may play a role in modulating oncogenic pathways to impact the tumor microenvironment, immune response, and, ultimately, patient outcomes.

From a clinical standpoint, PIEZO1 appears to be a dynamic ion channel protein, affecting varying cancers differently. Upregulated PIEZO1 is associated with worse overall survival outcomes and higher WHO grades in glioma patients [33]. On the other hand, previous studies attribute increased levels of PIEZO1 expression to better overall survival in non-small-cell lung cancer patients [34]. In breast cancer, studies have suggested that increased PIEZO1 expression leads to worse survival outcomes in certain subtypes, including lymph-node-positive, luminal A, and estrogen-receptor-positive patients, but have opposite results in other breast cancer subtypes, such as basal-like breast cancer [35]. To further understand the role of PIEZO1 in breast cancer, we investigated the correlation between PIEZO1 levels and clinical outcomes, and aimed to identify possible underlying mechanisms through a bioinformatic approach to add to our understanding concerning the PIEZO1 ion channel in breast cancer.

We employed TCGA’s provisional dataset, which contains transcriptional and clinical data from 1100 breast cancer patients. The use of these data enabled us to conduct a large-scale analysis of survival outcomes by PIEZO1 expression in breast cancer patients. The use of TCGA’s collection of clinical and genomic data from a large sample size ensured that the results were more comprehensive, less susceptible to variability, and more sufficiently powered. We utilized the Gene Expression Omnibus (GEO) datasets as validation cohorts for survival analysis by PIEZO1 expression and for exploring neoadjuvant chemotherapy responses by PIEZO1 expression. The Breast Cancer Gene-Expression Miner (bc-GenExMiner) was used as a tool for the analysis of data from many well-annotated transcriptomic datasets to obtain a comprehensive understanding of the correlation between PIEZO1 and prognostic indicators in breast cancer. Overall, the use of a bioinformatic approach using multiple publicly available datasets enabled us to make inferences regarding the role of PIEZO1 in breast cancer across a large set of patients.

## 2. Materials and Methods

### 2.1. Data Acquisition and Pre-Processing

As a discovery cohort, The Cancer Genome Atlas (TCGA) breast cancer cohort was used. There are 1100 patients in TCGA’s breast cancer cohort with clinical and gene expression data from RNA sequencing. Data were downloaded through cBioPortal (TCGA provisional dataset) [36,37]. Out of 1100, 1093 tumors were primary tumors. Of those, 1090 patients had overall survival data, and 999 patients had disease-free survival data. As a validation cohort, we used the GSE3494 breast cancer cohort from the Gene Expression Omnibus (GEO). Among 251 patients, 236 had disease-specific survival data. The patients were divided into PIEZO1 high and low expression groups with median cutoff. For drug response analyses, we used a breast cancer neoadjuvant cohort, GSE20271, GSE20194, and GSE25066. Since TCGA and GEO are de-identified publicly available cohorts, institutional review board approval was waived.

### 2.2. Gene Set Enrichment Analysis (GSEA)

GSEA was carried out comparing the PIEZO1 high and low groups using software provided by the Broad Institute (http://software.broadinstitute.org/gsea/index.jsp, accessed on 7 December 2020) using hallmark gene sets. False discovery rate (FDR) < 0.25 was considered significant.

### 2.3. Differentially Expressed Genes (DEG) and Pathway Enrichment Analysis

DEG analysis was conducted comparing the PIEZO1 high and low groups using the DEseq2 package version 3.18 (https://bioconductor.org/packages/release/bioc/html/DESeq2.html, accessed on 10 January 2024). We defined *p* < 0.01 and log2 fold change > |1| as significantly differentially genes and used them for pathway analysis. Pathway enrichment analysis was conducted using the KEGG pathway. Adjusted *p* < 0.05 was considered significant.

### 2.4. Breast Cancer Gene-Expression Miner v5.0 (bc-GenExMiner v5.0)

Breast Cancer Gene-Expression Miner version 5.0 (bc-GenExMiner v5.0) is an online statistical mining tool containing publicly available annotated breast cancer transcriptomic data, composed of both DNA microarrays (*n* = 11,552) and RNA-seq data (*n* = 5023). The “Expression” module analyses [38] were conducted to evaluate the PIEZO1 expression levels in breast cancers (from all DNA microarrays) with different clinicopathological characteristics and prognostic indicators. Dunnett–Tukey–Kramer’s test was used for this expression statistical analysis. The “Prognostic” module analyses [39] were used to generate survival curves for disease-free and overall survival based on PIEZO1 expression (from all RNA-seq) in different breast cancer cohorts. Survival statistical significance from bc-GenExMiner was determined using Cox Univariate analysis. Median was used as the cutoff, and *p* < 0.05 was considered statistically significant. 

### 2.5. Statistical Analysis

Survival analysis was conducted by Kaplan–Meier curve with log-rank test. The continuous value between two groups was compared by a Student’s *t*-test, and ANOVA followed by post hoc Tukey was used for the comparison of more than two groups. The cell composition fraction of the tumor was estimated by the CIBERSORT algorithm [40] and compared by a Wilcoxon test. Two-sided *p* < 0.05 was considered statistically significant for all tests unless otherwise stated. All statistical analyses were performed using R software (http://www.r-project.org/, version 4.3.2) together with Bioconductor (http://bioconductor.org/, version 3.18).

## 3. Results

### 3.1. Hormone-Receptor (HR)-Negative Patients Have Increased PIEZO1 Expression

First, we examined the PIEZO1 expression levels in primary tumors among different subtypes of breast cancer in the TCGA cohort. Based on estrogen receptor (ER) and progesterone receptor (PgR), the patients were divided into either hormone-receptor (HR)-positive (ER-positive, PgR-positive, or positive for both) or HR-negative (ER-negative and PgR-negative) groups. The patients were also divided into groups based on their human epidermal growth factor receptor 2 (HER2) status. We found that TNBC, which is HR-negative and HER2-negative, had significantly higher levels of PIEZO1 expression than both HR-positive HER2-negative (*p* < 0.0001) and HR-positive HER2-positive subtypes (*p* = 0.022) (Figure 1A). Upon the analysis of patients with breast tumors classified by HR status (Figure 1B) and HER2 status (Figure 1C), we found that the HR-negative patients had significantly higher levels of PIEZO1 expression than the HR-positive patients (*p* < 0.0001) (Figure 1B). There was no statistically significant difference in PIEZO1 expression between the HER2-negative and -positive groups (Figure 1C). These findings were validated through the analysis of an independent cohort, GSE3494, where the HR-negative cohort had significantly higher PIEZO1 expression than HR-positive patients (*p* = 0.006) (Figure 1D).

Next, we investigated whether there were differences in PIEZO1 expression in primary tumors between different American Joint Committee on Cancer (AJCC) stages in the whole breast cancer cohort. As shown in Figure 1E, there were no statistically significant differences in PIEZO1 expression between any of the clinical stages within the whole cohort. There was also no significant difference in PIEZO1 expression in relation to pathological T classification (pT), reflecting primary tumor sizes (Appendix A). Similarly, there was no significant difference in PIEZO1 expression in relation to pathological N classification (pN), reflecting the level of tumor spread to lymph nodes (Appendix A), nor based on pathological M classification (pM), reflecting metastasis status (Appendix A). Collectively, our results demonstrate that HR-negative breast cancer subtypes have increased PIEZO1 expression compared to HR-positive subtypes.

### 3.2. HR-Negative Breast Cancer Patients with High PIEZO1 Tumors Exhibit Worse Survival Outcomes

Next, we explored whether PIEZO1 level is associated with breast cancer patient survival. The breast cancer patients of the whole TCGA cohort were divided into high or low PIEZO1 groups based on a median cutoff. There was no significant difference in disease-free (Figure 2A) nor overall survival (Figure 2D) based on PIEZO1 expression in the whole cohort. Stemming from our results of differential PIEZO1 expression levels in different subtypes of breast cancer, we then divided the TCGA cohort into breast cancer subtypes characterized by HR status and HER2 status to examine survival. High-PIEZO1 breast cancer patients exhibited significantly worse disease-free survival (*p* = 0.033) and overall survival (*p* = 0.002) compared to the low-PIEZO1 group in the HR-negative cohort, while there was no significant difference in disease-free nor overall survival between the high- and low-PIEZO1 groups in the HR-positive cohort (Figure 2B,E). There was no statistically significant difference in disease-free nor overall survival outcomes between the high- and low-PIEZO1 groups in the HER2-positive subtype (Figure 2C,F). In the HER2-negative cohort, high PIEZO1 was correlated with worse disease-free survival (*p* = 0.046) (Figure 2C); however, there was no statistically significant difference in overall survival according to PIEZO1 levels in the HER2-negative subtype (Figure 2F).

To validate our results from TCGA, we utilized another independent cohort, GSE3494. Upon evaluating the whole cohort of GSE3494, there was a significant difference in disease-specific survival between the high- and low-PIEZO1 groups, with the high-PIEZO1 group having worse disease-specific survival (*p* = 0.01) (Figure 3A). Within the HR-negative cohort of GSE3494, patients with high PIEZO1 expression exhibited worse disease-specific survival compared to the low-PIEZO1 group (*p* = 0.003) (Figure 3C), whereas there was no difference in terms of disease-specific survival in the HR-positive cohort (Figure 3B). For further validation of these results, we utilized bc-GenExMiner for survival analysis. We saw similar results, with high-PIEZO1 HR-negative cohorts having worse overall survival outcomes (*p* = 0.0152), while there was no difference in overall survival according to PIEZO1 expression within the HR-positive cohort (*p* = 0.5014) (Appendix A). Therefore, our results highlight that high PIEZO1 expression levels correlate with worse survival outcomes in HR-negative breast cancer patients.

We further validated our results of PIEZO1 as a prognostic factor by investigating PIEZO1 expression in correlation with clinical prognostic indexes. Based on previous literature investigating prognostic indicators in breast cancer [35,41], we used bc-GenExMiner whole-cohort data to determine the PIEZO1 expression levels throughout different values of the Nottingham Prognostics Index (NPI). Our results show that the NP2 and NP3 cohorts had higher PIEZO1 expression than NP1 (*p* < 0.0001) (Appendix A). We also analyzed the PIEZO1 expression in cohorts differentiated by Scarff–Bloom–Richardson (SBR) grade, which reflects tumor grade and predicts prognosis in breast cancer patients. Our results demonstrate that the PIEZO1 expression was highest in the SBR3 cohort and lowest in the SBR1 cohort (*p* < 0.0001) (Appendix A). These results further validated our results of a correlation between unfavorable prognostic indicators and PIEZO1 expression.

We narrowed down the HR-negative patients into TNBC and HR-negative HER2-positive cohorts in the TCGA cohort. We saw worse overall survival (*p* = 0.014) and disease-free survival (*p* = 0.034) in high-PIEZO1 TNBC patients, but there were no significant differences in survival in HR-negative HER2-positive patients according to PIEZO1 expression (Appendix A). Though the HR-negative HER2-positive group had a low patient number, the survival curve did not separate, which implies that PIEZO1 did not predict survival in the HR-negative and HER2-positive cohort (Appendix A). These results imply that PIEZO1 shows prognostic value in HR-negative breast cancer, especially TNBC patients.

We investigated whether the PIEZO1 prognostic value in HR-negative breast cancer enhances with clinical characteristics such as stage, menopause status, and race in the TCGA cohort. First, we examined survival outcomes by PIEZO1 expression between those with Stage I/II and Stage III/IV HR-negative breast cancer. There was no significant difference in the PIEZO1 expression between Stage I/II and III/IV patients (Appendix A). Both Stage I/II (*p* = 0.02) and Stage III/IV (*p* = 0.033) patients with high PIEZO1 expression had worse overall survival outcomes than those in the low-PIEZO1 groups (Appendix A). While disease-free survival showed similar trends, the difference did not reach statistical significance (Appendix A). We investigated whether there were differences in PEIZO1 expression based on the menopausal status and race of the HR-negative patients. We saw no statistically significant difference in PIEZO1 expression between pre-menopausal and post-menopausal patients (Appendix A). Interestingly, in the survival analysis, we only observed a difference in overall survival in post-menopausal HR-negative patients, where high PIEZO1 patients had worse overall survival outcomes (*p* = 0.037) (Appendix A). There was also a trend of worse disease-free survival outcomes in the high-PIEZO1-expression patients of the post-menopausal cohort, although it did not reach statistical significance, which may be due to a low patient number (Appendix A). Therefore, PIEZO1 may enhance its prognostic potential in post-menopausal HR-negative patients. When comparing Caucasian patients (White) with African American patients (Black) with HR-negative breast cancer, interestingly, our results demonstrated that Black patients had higher PIEZO1 expression levels than White patients (*p* = 0.002) (Appendix A). There was also a trend of worse survival outcomes in high-PIEZO1-expression patients, although it did not reach statistical significance (Appendix A).

### 3.3. PIEZO1 Did Not Predict Chemotherapy Response in HR-Negative Tumors

To determine whether PIEZO1 expression predicts neoadjuvant chemotherapy outcomes in HR-negative patients, we utilized three independent datasets—GSE20271, GSE20194, and GSE25006 (ER-negative). We observed no significant differences in PIEZO1 expression in pre-treatment tumors between those who exhibited complete pathological responses (pCR) and those who had residual disease (RD) in response to neoadjuvant chemotherapy treatment in HR-negative HER2-positive or TNBC patients (Appendix A) Our results demonstrate that, by itself, PIEZO1 expression was not sufficient to correlate with response to neoadjuvant chemotherapy (Appendix A).

### 3.4. EMT and Multiple Pro-Tumor Signaling Pathways Are Enriched in High-PIEZO1 HR-Negative Tumors

To investigate the possible underlying mechanisms of how high PIEZO1 expression contributes to worse survival outcomes within HR-negative breast cancer cohorts, we performed Gene Set Enrichment Analysis (GSEA) comparing high- and low-PIEZO1 HR-negative tumors of the TCGA dataset. The gene sets significantly enriched in high-PIEZO1 HR-negative tumors are depicted in Figure 4. We found that EMT was the most enriched pathway within high-PIEZO1 HR-negative tumors. Other gene sets related to cancer aggressiveness, including hypoxia, apical junction, angiogenesis, and glycolysis, were also enriched in high-PIEZO1 HR-negative tumors. Additionally, multiple signaling pathways, including TGF-β1, TNF-α via NF-κB, and WNT/β-catenin signaling, were enriched in the high-PIEZO1 HR-negative tumors. These results suggest that PIEZO1 may play a role in promoting a hypoxic and glycolytic tumor environment with increased EMT and angiogenesis driven by TGF-β1, TNF-α via NF-κB, and WNT/β-catenin signaling to promote more aggressive features in high-PIEZO1 HR-negative breast tumors.

To further characterize the pathways that are positively or negatively associated with PIEZO1 expression in HR-negative breast tumors of TCGA, we identified differentially expressed genes (DEGs) that are upregulated or downregulated in association with PIEZO1 levels. We identified 205 upregulated and 147 downregulated DEGs (Figure 5A). The pathway analysis of these DEGs showed that the “ECM–receptor interaction” and “Focal Adhesion” pathways were highly enriched in high-PIEZO1 HR-negative tumors (Figure 5B). These results suggest that PIEZO1 has a direct correlation with modulating the cellular junctions and cytoskeleton composition. On the other hand, the “Neuroactive ligand–receptor interaction” pathway was enriched in the low-PIEZO1 HR-negative tumors (Figure 5C). The identification of these enriched pathways, especially those that are enriched in high-PIEZO1 tumors, provides more insight into the potential roles of PIEZO1 within HR-negative breast cancer.

### 3.5. Decreased T Cell Infiltration Exhibited in High-PIEZO1 HR-Negative Tumors

To further explore the possible mechanisms of how PIEZO1 may contribute to worse clinical responses in HR-negative breast cancer, we investigated whether there is an association between PIEZO1 and the anti-cancer immune response in HR-negative breast cancer. To do so, we utilized a CIBERSORT algorithm to estimate the immune cell infiltration in HR-negative primary tumors. Our results showed that patients characterized by high PIEZO1 expression had decreased levels of CD8+ T cells within their tumors (*p* = 0.002) when compared to patients with low PIEZO1 expression (Figure 6). Similarly, patients with high PIEZO1 had lower levels of activated CD4+ memory T cells within the tumors than those with low PIEZO1 expression (*p* = 0.020) (Figure 6). There was no statistically significant difference in infiltration level by PIEZO1 expression for other immune cell types. These results suggest that high-PIEZO1 HR-negative tumors have a suppressed anti-tumor immune response in terms of T cell infiltration in comparison to low-PIEZO1 HR-negative tumors.

## 4. Discussion

To gain insight into the role of mechanosensation in breast cancer, we investigated the role of the PIEZO1 mechanosensory ion channel on breast cancer survival. Our results demonstrate that high-PIEZO1 patients have worse survival outcomes than low-PIEZO1 patients, specifically in HR-negative and TNBC cohorts. These results are clinically significant in that they suggest PIEZO1 could be utilized as a prognostic biomarker in HR-negative breast cancer patients to identify patients at an increased risk for mortality from HR-negative breast cancers. This is supported by the NPI and SBR analysis, where the PIEZO1 expression was elevated in higher-grade tumors. We did not observe a difference in survival based on the PIEZO1 expression level between HER2-positive and -negative patients. The survival outcomes in HR-negative patients based on PIEZO1 expression were consistent between the TCGA, GSE3494, and bc-GenExMiner datasets, whereas there was a discrepancy in the statistically significant survival outcomes of the whole cohort between the TCGA and GSE3494 datasets, although the trends were the same. The significant difference in disease-specific survival between high- and low-PIEZO1 patients observed in the whole cohort in GSE3494 may be due to the difference in the patient population and differing methods to define ER/PgR status between datasets and follow up. We also demonstrated that HR-negative subtypes, which are known to be more aggressive than HR-positive subtypes [42], have increased levels of PIEZO1 expression compared to the HR-positive subtype. Therefore, our results suggest that PIEZO1 primarily contributes to worse survival outcomes in HR-negative patients and specifically in TNBC, implying that PIEZO1 may have different effects in different subtypes of breast cancer. The idea that PIEZO1 has differential functions in varying subtypes of breast cancer is further supported by previous literature on PIEZO1 in breast cancer. Chen et al. [41] and Xu et al. [35] reported that high PIEZO1 correlated with worse survival outcomes in many breast cancer subtypes, but better outcomes in basal-like breast cancer, while our results demonstrate that the worse outcomes associated with high PIEZO1 are primarily in the HR-negative and TNBC subtypes; this highlights the ambiguity of PIEZO1 on survival outcomes amongst breast cancer subtypes. Despite these findings, we did not demonstrate differences in response to chemotherapy according to PIEZO1 expression in HR-negative patients. Further investigation into the function of PIEZO1 in breast cancer subtypes is necessary.

As a possible mechanism, our results demonstrate that increased PIEZO1 expression levels are correlated with the upregulation of gene sets associated with cancer aggressiveness, including EMT, hypoxia, apical junction, angiogenesis, and glycolysis with multiple signaling pathways, including TGF-β1, TNF-α via NF-κB, and WNT/β-catenin, in high-PIEZO1 HR-negative tumors. Our results are consistent with previous studies that implicate PIEZO1 participation in EMT [14,22], hypoxia-response gene expression [18,22], angiogenesis [16], glycolytic metabolism [19], and the multiple signaling pathways [14,16,20,21,34]. EMT, hypoxia, and angiogenesis are cancer-promoting and are well characterized to be associated with worse survival outcomes in cancer patients [43,44,45]; thus, the enrichment of these processes in high-PIEZO1 HR-negative tumors may indicate the mechanistic roles of PIEZO1 in contributing to the worse survival outcomes of these patients. We also showed that PIEZO1 was highly correlated with the formation of apical junction complex (AJC) proteins, ECM ligand–receptor interactions, and focal adhesion pathways. The enrichment of the AJC gene set likely relates to EMT, which was the most highly enriched gene set in high-PIEZO1 HR-negative tumors. AJC protein deregulation occurs in EMT to induce the loss of apical–basal polarity and cytoskeletal remodeling, which are required for the acquisition of mesenchymal and invasive characteristics [46]. Our enrichment results suggest that the PIEZO1 ion channel may respond to mechanical forces to induce cellular junction deregulation, cytoskeletal remodeling, and TME modulation, thus promoting EMT and the motility of cancer cells. A recent study conducted by Luo et al. supports this hypothesis, as they observed that compression forces on breast cancer cells enhanced invasion capabilities in a PIEZO1-dependent manner [47]. Another study suggested that increased PIEZO1 expression correlated with the elevation of purine metabolism pathways [41], which have been demonstrated to induce EMT and cell migration [48]. Therefore, it would be interesting to further investigate the connections between PIEZO1 and cytoskeleton remodeling, cellular metabolism, and metastasis in breast cancer.

Our results also show that PIEZO1 plays a role in the immune cell infiltration into HR-negative tumors. There is an inverse correlation of PIEZO1 expression level to CD8+ and activated CD4+ memory T cell infiltration in HR-negative cohorts. Thus, PIEZO1 may play a role in CD8+ and CD4+ T cell exclusion from HR-negative tumors. As a mechanosensory protein, PIEZO1 may respond to external forces and participate in the modulation of the TME and cytoskeletal reorganization to establish an environment that physically inhibits CD8+ and CD4+ T cells from penetrating the tumor.

In addition, we showed that PIEZO1 is correlated with EMT, hypoxia, and TGF-β signaling, processes which have all been shown to be correlated with decreased CD8+ and/or CD4+ T cell infiltration in various cancer types [28,29,30]. PIEZO1 may participate in these processes to reduce T cell infiltration into the tumor. CD8+ T cells are major drivers of the anti-cancer immune response, utilizing granzymes and perforins to kill the target cancer cells, while CD4+ T cells contribute to the anti-cancer immune response through CD8+ T cell priming and the activation of innate immune cells [31,49]. In breast cancer, we have shown that higher levels of CD8+ T cell infiltration and increased CD4+ activity are correlated with a better patient prognosis [32]. Due to the importance of CD8+ and CD4+ T cells in the anti-cancer immune response, the exclusion of CD8+ T cells and activated CD4+ memory T cells from high-PIEZO1 HR-negative tumors enables cancer progression and likely contributes to the worse survival outcomes that we see in HR-negative breast cancer patients with high PIEZO1 expression levels. Interestingly, while we saw decreased levels of these effector cells in the high-PIEZO1 groups, this same observation was not seen for other immune cell types. Macrophages, natural killer (NK) cells, dendritic cells, and other immune cells showed no significant difference in infiltration levels between high- and low-PIEZO1 HR-negative tumors. As a mechanosensory ion channel, PIEZO1 may participate in the physical modulation of the TME to lead to immune cell exclusion, but our results suggest that this is not the only function of PIEZO1 within the HR-negative tumors. If this was the case, we would expect to see lower levels of all immune cell types in high-PIEZO1 tumors, rather than the exclusion of only CD8+ and CD4+ T cells. CD8+ and CD4+ T cells are well established as the most prominent mediators of anti-cancer immunity [50], so our results suggest that PIEZO1 may also have a molecular role that reflects cancer aggressiveness and the suppression of anti-tumor immunity in HR-negative breast cancer.

Our study, in conjunction with past studies, exemplifies the importance of mechano-signaling, and specifically the PIEZO1 ion channel, in the clinical outcomes of breast cancer patients. Decreased levels of CD8+ and CD4+ T cell infiltration, along with the enrichment of important cancerous processes like EMT, the modulation of cell–cell interactions, hypoxia, and angiogenesis, may explain why high-PIEZO1 patients within the HR-negative cohort exhibit worse survival outcomes than those with low PIEZO1 expression. Our findings of increased PIEZO1 expression in African American patients can be one of the contributing factors in disparities in outcomes in this patient population. Further research is needed to determine how PIEZO1 mechanistically acts to worsen patient prognosis in HR-negative cohorts and TNBC patients, and whether PIEZO1 could be utilized as a viable therapeutic target.

This study has limitations. First, this was an observational study based on bioinformatic analyses in publicly available patient data with known bias. To validate our bioinformatic results, we are conducting further experimental studies to identify the mechanistic role of PIEZO1 and confirm the involvement of PIEZO1 in highlighted pro-cancer pathways and T cell infiltration. Another caveat is that we analyzed primary tumors only, so the role of PIEZO1 in metastatic sites is undetermined.

## 5. Conclusions

Our study demonstrated that HR-negative breast cancer patients have higher PIEZO1 ion channel expression levels than HR-positive patients, and that high PIEZO1 expression is attributed to worse survival in HR-negative cohorts, especially in TNBC patients. Our study suggests that worse survival outcomes in high-PIEZO1 HR-negative patients may be due to the promotion of EMT, hypoxia, angiogenesis, and glycolysis through multiple signaling pathways such as TGF-β1, TNF-α via NF-κB, and WNT/β-catenin signaling. It may also be due to the PIEZO1-mediated modulation of the TME, resulting in the exclusion of CD8+ and CD4+ T cells from HR-negative tumors. Our findings demonstrate the clinical relevance of the PIEZO1 ion channel in breast cancer and highlight the importance of PIEZO1 expression on patient survival and its potential as a prognostic biomarker in HR-negative breast cancer patients. Further studies investigating the mechanistic roles of PIEZO1 in breast cancer are warranted.

## Figures and Tables

**Figure 1 cancers-16-00683-f001:**
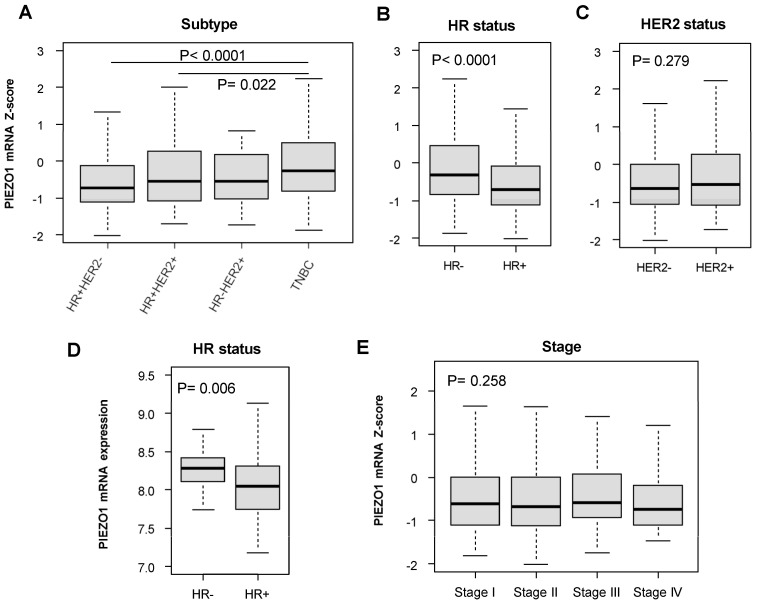
PIEZO1 expression and subtype. (**A**) PIEZO1 expression levels by hormone receptor (HR) status and HER2 status in TCGA breast cancer cohort. HR+HER2−: HR positive and HER2 negative, *n* = 606; HR+HER2+: HR positive and HER2 positive, *n* = 146; HR-HER2+: HR negative and HER2 positive, *n* = 38, TNBC: Triple-negative breast cancer, *n* = 160. (**B**) PIEZO1 expression levels by HR status in TCGA breast cancer cohort. HR−: HR negative, *n* = 219; HR+: HR positive, *n* = 823. (**C**) PIEZO1 expression levels by HER2 status in TCGA breast cancer cohort. HER2−: HER2 negative, *n* = 768; HER2+: HER2 positive, *n* = 185. (**D**) PIEZO1 expression by HR status in GSE3494. HR−: HR negative, *n* = 34; HR+: HR positive, *n* = 213. (**E**) PIEZO1 expression levels in stages of AJCC TNM system in TCGA breast cancer cohort. Stage I: *n* = 181; Stage II: *n* = 619; Stage III: *n* = 249; Stage IV: *n* = 20. Continuous value between two groups was compared by Student’s *t*-test, and ANOVA followed by post hoc Tukey was used for comparison of more than two groups.

**Figure 2 cancers-16-00683-f002:**
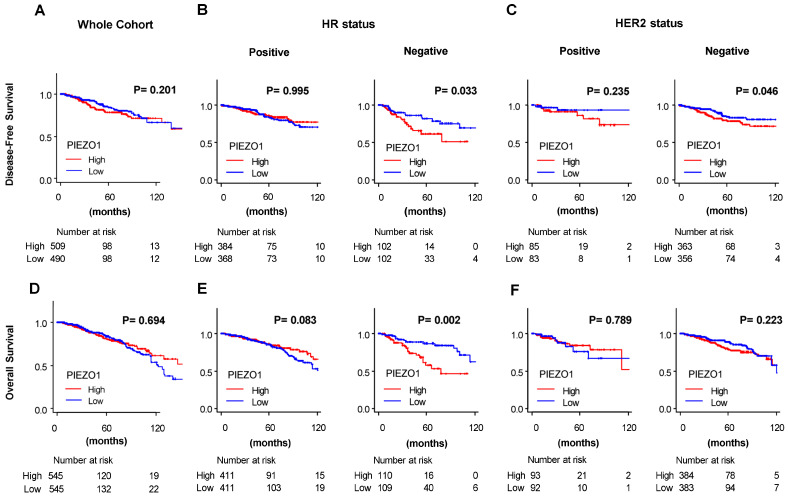
Breast cancer patient survival by PIEZO1 expression in TCGA. (**A**) Disease-free survival by PIEZO1 expression in the whole cohort, (**B**) in the hormone receptor (HR) positive and negative cohort, and (**C**) in the HER2 positive and negative cohort. (**D**) Overall survival by PIEZO1 expression in the whole cohort, (**E**) in the hormone receptor (HR) positive and negative cohort, and (**F**) in the HER2 positive and negative cohort. Survival difference was estimated by log-rank test.

**Figure 3 cancers-16-00683-f003:**
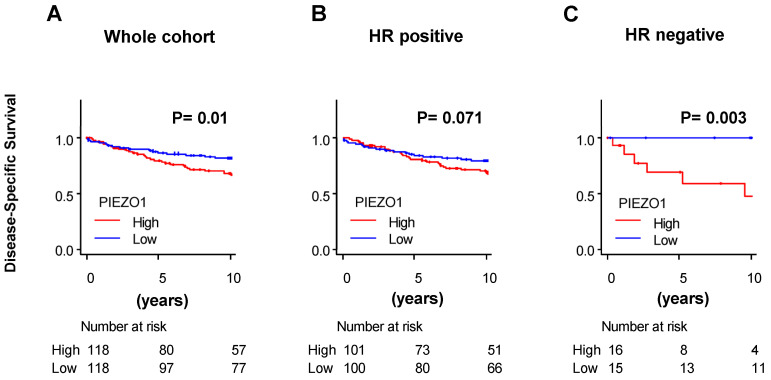
PIEZO1 expression in breast cancer patients in GSE3494. (**A**) Disease-specific survival by PIEZO1 expression in whole cohort, (**B**) in the hormone receptor (HR) positive and (**C**) negative cohort of GSE3494. Survival difference was estimated by log-rank test.

**Figure 4 cancers-16-00683-f004:**
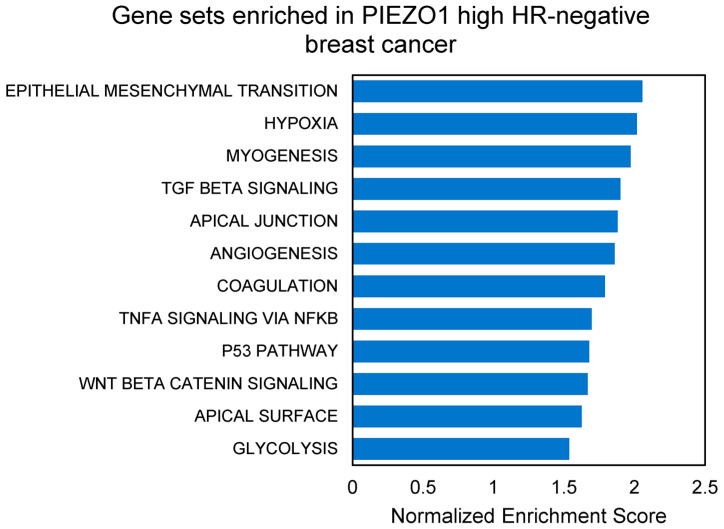
Significantly enriched gene sets in high-PIEZO1 HR-negative breast tumors of TCGA cohort. Gene sets identified by Gene Set Enrichment Analysis (GSEA) using hallmark gene sets with false discovery rate (FDR) < 0.25.

**Figure 5 cancers-16-00683-f005:**
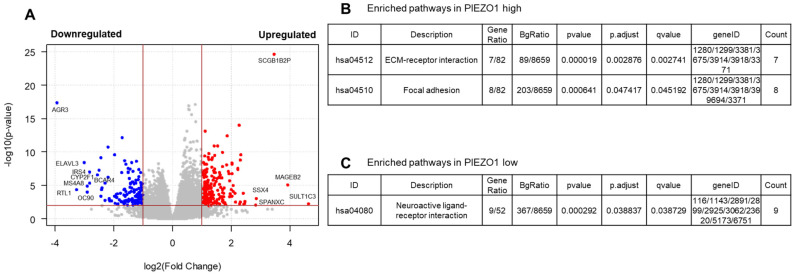
Differentially expressed genes (DEGs) and pathways by PIEZO1 expression in TCGA HR-negative breast cancer cohort. (**A**) Volcano plot of DEGs. Red (right) panel exhibits upregulated genes in association with PIEZO1 expression. Blue (left) panel exhibits downregulated genes in association with PIEZO1 expression. The cutoffs of *p* < 0.01 and log2 fold change > |1| were used to identify significantly differentially genes. Pathways enriched in (**B**) high-PIEZO1 and (**C**) in low-PIEZO1 HR-negative tumors. KEGG pathway used for pathway enrichment, with adjusted *p* < 0.05 being significant.

**Figure 6 cancers-16-00683-f006:**
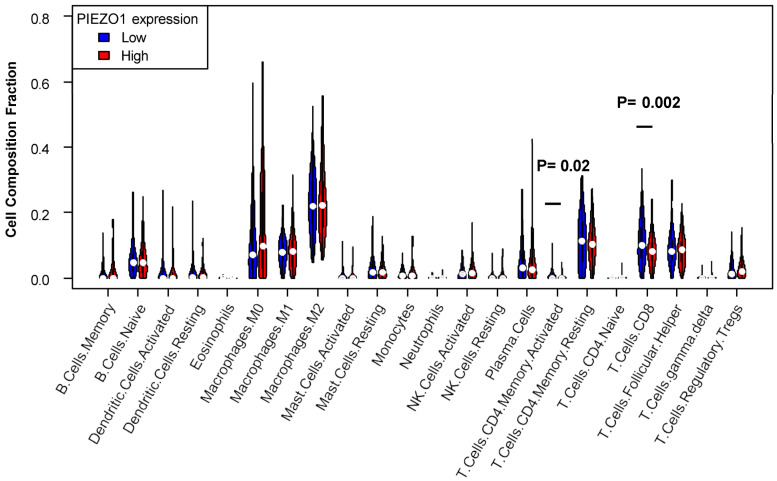
Infiltrated immune cell estimated by CIBERSORT in HR-negative breast cancer by PIEZO1 expression in TCGA. Cell composition fraction of the primary tumor was estimated by CIBERSORT algorithm and compared by Wilcoxon test.

## Data Availability

TCGA provisional database was downloaded through cBioportal (http://www.cbioportal.org/, accessed on 1 December 2020) and GSE3494, GSE20271, GSE20194, and GSE25066 were downloaded through Gene Expression Omnibus (GEO; https://www.ncbi.nlm.nih.gov/geo/, accessed on 1 December 2020).

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
