# Peer review of "Increased PIEZO1 Expression Is Associated with Worse Clinical Outcomes in Hormone-Receptor-Negative Breast Cancer Patients"

_cancers, 2024, doi:10.3390/cancers16040683_

Round 1

Reviewer 1 Report

Comments and Suggestions for Authors

I am very sceptical about articles in which the authors only analyse available databases or the Cancer Genome Atlas and GSE3494 and draw conclusions about the survival of patient groups!!! The lack of own clinical material is my first major concern. The analysis of the atlas data is good, but only as a complementary analysis to the authors' clinical patients material. Furthermore, why did the authors not perform any experiments with cell lines?
The introduction is very poor and the presentation of such a complex problem as breast cancer is unacceptable.
The paper needs a lot of improvement, including analysis of clinical material from patients and even simple in vitro tests with cell lines.

Author Response

Please see the attached response document.

Reviewer 2 Report

Comments and Suggestions for Authors

The manuscript investigates the role of the PIEZO1 ion channel in breast cancer, with a focus on its association with survival outcomes and potential mechanisms in different breast cancer subtypes. The study utilizes data from The Cancer Genome Atlas (TCGA) and Gene Expression Omnibus (GEO) cohorts to explore PIEZO1 expression patterns and their impact on clinical outcomes. The study provides valuable insights into the clinical relevance of PIEZO1 in HR-  breast cancer, underscoring the potential of PIEZO1 as a prognostic biomarker and highlighting its role in modulating key pathways and the immune microenvironment. To provide a more comprehensive understanding, the following analysis could be incorporated:

1) The findings solely rely on bioinformatic analyses of publicly available data. Although the manuscript mentions ongoing experimental studies for validation, the lack of experimental data in the current manuscript limits the strength of the conclusions.

2) The findings are specific to HR- breast cancer subtypes. Have the authors examined PIEZO1 expression in HR-HER+ and TNBC subtypes?

3) Could the integration of patient cohort clinical characteristics enhance the prognostic potential of PIEZO1 as a biomarker?

4) Besides GSEA, differential expression analysis could be conducted to identify genes that are significantly upregulated or downregulated in association with PIEZO1 expression.

5) Explore pathway enrichment analysis beyond GSEA to identify pathways associated with differentially expressed genes.

6) Any available drug response data to identify potential therapeutic options or drug resistance patterns associated with PIEZO1 expression.

7) Any available Single-Cell RNA Sequencing (scRNA-seq) dataset for analysis to explore PIEZO1 expression at the single-cell level.

Author Response

Please see attached response document.

Reviewer 3 Report

Comments and Suggestions for Authors

The manuscript utilizes bioinformatics analyses of datasets from The Cancer Genome Atlas (TCGA) and GSE3494 to investigate the correlation between PIEZO1 expression and survival outcomes. It presents the compelling finding that high PIEZO1 expression correlates with poorer survival in hormone receptor-negative breast cancer patients and is linked to aggressive cancer pathways, including epithelial-mesenchymal transition, hypoxia, and reduced T cell infiltration. The bioinformatic analyses conducted are robust, and the conclusions drawn are persuasive. However, I have several comments that I believe the authors should address to enhance the clarity and completeness of the study:

Major Comments:

·         The focus of this study on PIEZO1 expression in hormone receptor-negative breast cancer and its connection with clinical outcomes is timely and relevant. The manuscript aligns well with existing literature that underscores PIEZO1's role in breast cancer, highlighting its importance in disease progression and its potential as a therapeutic target or prognostic biomarker. However, the manuscript does not reference two critical studies that previously reported on the prognostic values of PIEZO1 in breast cancer (PMID: 33559992; 35351580). I recommend that the authors incorporate these references into their discussion. This would not only contextualize their findings within the broader research landscape but also allow them to underscore the novel contributions of their study compared to these previous works.

·         The findings of this manuscript are significant, contributing to the expanding understanding of PIEZO1's role in breast cancer. However, it's important to recognize that the association of PIEZO1 with poor prognosis in breast cancer, particularly in terms of its influence on cancer cell behavior and mechanotransduction, has been previously documented. In light of this, I suggest that the authors more explicitly acknowledge these prior studies. Doing so would help situate their findings within the existing body of research and underscore any unique aspects of their study, such as specific methodological approaches or analytical insights.

·         For enhanced clarity and replicability, it is essential that P-values and the specific tests employed are clearly described in the legends of each figure. This level of detail is crucial for the scientific community to fully appreciate and evaluate the robustness of the findings presented.

Overall, this study makes a valuable contribution to the field of breast cancer research. Addressing these points will strengthen the manuscript and more clearly delineate its contributions to the current understanding of PIEZO1 in hormone receptor-negative breast cancer.

Comments on the Quality of English Language

N/A

Author Response

(The authors gave the same response as above.)

Round 2

Reviewer 1 Report

Comments and Suggestions for Authors

Please describe these explanations about the use of database (in great detail) in the introduction of your paper.
